# Association of Serum Uric Acid with Cardiovascular Disease Risk Scores in Koreans

**DOI:** 10.3390/ijerph16234632

**Published:** 2019-11-21

**Authors:** Seung Yun Lee, Won Park, Young Ju Suh, Mie Jin Lim, Seong-Ryul Kwon, Joo-Hyun Lee, Young Bin Joo, Youn-Kyung Oh, Kyong-Hee Jung

**Affiliations:** 1Division of Rheumatology, Department of Internal Medicine, College of Medicine, Inha University, Incheon 22332, Korea; sylee0203@inhauh.com (S.Y.L.); parkwon@inha.ac.kr (W.P.); miejinl@inha.ac.kr (M.J.L.); rhksr@inha.ac.kr (S.-R.K.); 2Department of Biomedical Sciences, College of Medicine, Inha University, Incheon 22332, Korea; ysuh@inha.ac.kr (Y.J.S.); ykoh12@korea.ac.kr (Y.-K.O.); 3Department of Rheumatology, Inje University Ilsan Paik Hospital, Goyang 10380, Korea; jhl9325@gmail.com; 4Division of Rheumatology, Department of Internal Medicine, St. Vincent’s Hospital, College of Medicine, The Catholic University of Korea, Suwon 16247, Korea; addmedic@hanmail.net

**Keywords:** uric acid, cardiovascular disease

## Abstract

As the prevalence of gout and hyperuricemia increases, the comorbidities of gout and hyperuricemia have become a public health burden. In particular, risks of cardiovascular disease (CVD)-related complications are increasing. However, a few guidelines exist for the management of hyperuricemia. This cross-sectional study aimed to investigate the association of serum uric acid with CVD risk in the general population of Korean adults. We examined cross-sectional data from the first and second years of the seventh Korea National Health and Nutrition Examination Survey 2016–2017. Among 16,277 participants, 8781 were analyzed. We estimated the CVD risk using a 10-year CVD risk score prediction formula. There was a significant association of serum uric acid with 10-year CVD risk scores after adjusting for physical activity, body mass index, serum creatinine, and alcohol consumption in both sexes (*p* < 0.001). In the fitted fractional polynomial model, an approximate U-shaped association between serum uric acid levels and 10-year CVD risk scores was found in men. At the serum uric acid level of 6.9 mg/dL, the CVD risk was lowest. An approximate J-shaped association between serum uric acid levels and 10-year CVD risk scores was found in women. Our study showed that hyperuricemia was associated with an increased CVD risk. Hypouricemia was also associated with an increased CVD risk in men. We, therefore, recommend proper management of uric acid levels in the general population to reduce CVD risks.

## 1. Introduction

Gout is common inflammatory arthritis affecting >1% of the population in most developed countries [1]. Gouty arthritis is the representative complication of sustained hyperuricemia [2]. There is a tendency to increase substantially in the prevalence of gout and hyperuricemia [3]. Prevalence of gout and hyperuricemia in the United States in 2007−2008 were 3.9% and 21.4%, indicating increases of 1.2% and 3.2%, respectively, compared to in 1988−1994. The prevalence of comorbidities, including cardiovascular disease (CVD), hypertension, diabetes, and metabolic syndrome, was reported to be 2–3 times higher among patients with gout than among those without gout [4,5,6]. A high prevalence of comorbidities was found among patients with hyperuricemia. As the prevalence of gout and hyperuricemia increase in parallel, the comorbidities of gout and hyperuricemia created a public health burden, especially increasing CVD-related overall mortality [1].

CVD is the leading cause of death worldwide. In 2017, 17.8 million people died of CVD, representing 31.8% of all global deaths that year [7]. Recently there have been some studies about the association between serum uric acid level and CVD risk [8,9,10,11,12,13]. However, the nature of the association was controversial; moreover, no study has analyzed nationwide data. The 10-year CVD risk score based on the Framingham Heart Study is known for the most reliable tool to predict CVD using relatively simple methods [14]. This risk score had never been used in studies that investigated the relationship of serum uric acid level and CVD risk.

Here, we aimed to investigate the association between serum uric acid and CVD risk in a nationwide survey of the general population of Korean adults using the 10-year CVD risk score.

## 2. Materials and Methods

### 2.1. Data Source and Study Population

In this cross-sectional study, we examined data from a survey conducted during the first and second years of the seventh Korea National Health and Nutrition Examination Survey (KNHANES) 2016–2017. The KNHANES was conducted by the Korean Centers for Disease Control and Prevention and the Ministry of Health and Welfare. This nationwide survey was cross-sectional and representative of the non-institutionalized Korean population with a stratified multistage and clustered probability sampling method. The survey was composed of a health interview, nutrition assessment, and health examination. Detailed methods for the KNHANES data collection and the survey data can be viewed publicly on the KNHANES website (available at https://knhanes.cdc.go.kr/knhanes/eng/index.do). A total of 16,277 individuals, among which 21,236 eligible individuals participated in the KNHANES 2016–2017, and the response rate was 76.6%.

The participants in the present study were stratified into two groups depending on the serum uric acid level. Hyperuricemia was defined as a serum uric acid level of >7.0 mg/dL in men and >6.0 mg/dL in women [15]. We included participants aged 30–74 years without prevalent CVD based on the criteria from the Framingham Heart Study [14]. We excluded the following participants from the study population—individuals with current myocardial infarction, angina, or cerebral stroke, which are composites of CVD; those with cancer of the stomach, liver, colon, breast, uterine cervix, lung, thyroid, or other organs; those with current pulmonary tuberculosis; pregnant women; and participants with missing data. Finally, we analyzed a total of 8781 participants (Figure 1).

Informed consent for the survey was obtained from all the participants, and all the study processes were performed in accordance with the guidelines of the Strengthening the Reporting of Observational Studies in Epidemiology. The study protocol was approved by the institutional review board of Inha University Hospital (no. 2019−05−027), and complied with the principles of the Declaration of Helsinki.

### 2.2. General Characteristics, Comorbidities, and Health Behaviors

Data on demographic and socioeconomic characteristics and comorbidities were collected through personal interviews by trained personnel. Data on health behaviors were collected using self-reported questionnaires. Hypertension was defined as receiving current treatment for hypertension. Diabetes was defined as a diabetes diagnosis by a medical doctor. Smoking was categorized into two groups based on the classification of the Framingham Heart Study [14]. Alcohol consumption was categorized into two groups according to the standards of 70 g per occasion twice a week for men and 50 g per occasion twice a week for women.

### 2.3. Anthropometric Measurements

Weight and height were measured with the participants wearing light clothes without shoes. Blood pressure measurements were taken three times on the right arm after the participants had rested for at least 5 min in the sitting position.

### 2.4. Biochemical Measurements

Before blood samples were taken for biochemical measurements, all the participants fasted for at least 8 hours. The samples were refrigerated immediately, transported to the central testing institute, and analyzed within 24 hours. The serum uric acid level was measured using colorimetry with the uricase-peroxidase mechanism (Hitachi Automatic Analyzer 7600-210; Hitachi, Tokyo, Japan). Total cholesterol, triglyceride, high-density lipoprotein cholesterol (HDL-C), low-density lipoprotein cholesterol (LDL-C), and serum glucose were analyzed using the Hitachi Automatic Analyzer 7600-210. The serum creatinine level was measured using the compensated rate-blanked Jaffe assay (Roche, Mannheim, Germany) with the Hitachi Automatic Analyzer 7600-210.

### 2.5. Assessment of CVD Risk Based on a 10-Year CVD Risk Score

We estimated the CVD risk of the study population by using a 10-year CVD risk score prediction formula. The formula was developed using a sex-specific multivariate risk factor algorithm based on the Framingham Heart Study [14]. This scoring model is intended for participants free of CVD in a baseline examination and aged 30–74 years. This formula has been validated in large-population studies and allows the screening of candidates at high risk of initial CVD events among the general population. The 10-year CVD risk score is the calculated likelihood of CVD after 10 years based on age, total cholesterol, HDL-C, systolic blood pressure, hypertension treatment status, smoking status, and diabetes.

### 2.6. Assessment of Physical Activity

To evaluate the practice rate of physical activity, information about aerobic physical activity and muscular strength exercise was surveyed using interviewer-administered questionnaires. The Global Physical Activity Questionnaire, which was developed by the World Health Organization (WHO) for physical activity surveillance at a population level, was used to measure aerobic physical activity status [16,17]. The vigorous intensity was defined as the status of being out of breath heavily and continuously or having a very fast heart rate for ≥10 min. Moderate intensity was defined as the status of being out of breath mildly and continuously or having a slightly fast heart rate for ≥10 min. The participants reported their exercise frequency and duration in a typical week. The practice rate of aerobic physical activity was defined as performing moderate-intensity physical activity for ≥150 min per week or performing vigorous-intensity physical activity for ≥75 min per week or an equivalent combination of physical activity intensity based on the national guideline for physical activity [18]. The practice rate of muscular strength exercise was defined as performing strength exercises ≥2 times per week based on the national guideline for physical activity [18]. Muscular strength exercises included press-up, sit-up, dumbbell, barbell, and horizontal bar exercises. The participants reported their exercise frequency in the most recent week.

### 2.7. Statistical Analyses

To increase the representativeness and accuracy of the estimates, statistical procedures were performed to reflect the complex sampling design and sampling weights of the KNHANES. Continuous variables are presented as weighted means with standard deviations, while categorical variables are shown as weighted percentages. General characteristics were compared using the *t*-test for continuous variables and the chi-square test for categorical variables. The fractional polynomial analysis was performed to estimate the proper nonlinear functional relationship of serum uric acid level to the 10-year CVD risk. After the adjustment for serum uric acid, aerobic physical activity, muscular strength exercise, body mass index (BMI), serum creatinine, and alcohol consumption, the *β*-coefficients between these variables, and the 10-year CVD risk scores were calculated. Statistical analyses were performed using SAS version 9.3 (SAS Institute Inc., Cary, NC, USA) and R version 3.5.2 (The R Foundation for Statistical Computing; https://www.r-project.org). Two-sided *p* values of <0.05 were considered statistically significant.

## 3. Results

### 3.1. Participants’ Characteristics

The general characteristics of the study participants by sex and serum uric acid level are presented in Table 1. Among men, those with high uric acid levels were younger than those with low uric acid levels. However, in women, the result was the opposite regarding age. In men, education, alcohol consumption, diabetes status, systolic blood pressure, BMI, and levels of total cholesterol, HDL-C, and serum creatinine were significantly different depending on the serum uric acid level. In women, hypertension status, in addition to the aforementioned variables, significantly differed depending on the serum uric acid level.

### 3.2. Associations of Serum Uric Acid with CVD Risk

The *β*-coefficients between serum uric acid and covariates and the 10-year CVD risk score are presented in Table 2. The association between CVD risk and each factor, such as serum uric acid, aerobic physical activity, BMI, serum creatinine, and alcohol consumption, was statistically significant after adjusting for factors other than itself. In each group of men and women, the 10-year CVD risk score was negatively correlated with aerobic physical activity, but positively with BMI and serum creatinine. The 10-year CVD risk score was positively correlated with alcohol consumption in men, but negatively in women. There was no significant relationship between the 10-year CVD risk score and the muscular strength exercise. Figure 2 presents the association between serum uric acid level and the 10-year CVD risk score analyzed by fractional polynomial model. In the case of men, the third-degree equation cubic fraction was fitted as follows: 10-year CVD risk score = 0.023 × serum uric acid^3^ − 0.243 × serum uric acid^2^ – 1.067 × practice rate of aerobic physical activity + 0.452 × practice rate of muscular strength exercise + 0.144 × BMI + 0.763 × serum creatinine + 0.467 × alcohol consumption (root mean square error [RMSE] = 5.3714). The inflection point was estimated using a partial differential equation. An approximate U-shaped association between serum uric acid level and 10-year CVD risk score (predicted value) with an inflection point at 6.9 mg/dL serum uric acid level was found in men (Figure 2A). In women, the third-degree equation cubic fraction was fitted as follows: 10-year CVD risk score = 0.006 × serum uric acid^3^ − 0.827 × practice rate of aerobic physical activity − 0.012 × practice rate of muscular strength exercise + 0.517 × BMI + 2.776 × serum creatinine − 1.620 × alcohol consumption (RMSE = 5.4784). An approximate J-shaped association between serum uric acid level and 10-year CVD risk score was found in women (Figure 2B).

## 4. Discussion

In this nationwide survey of the general population, we found associations of serum uric acid with CVD risk. In men, both lower and higher serum uric acid levels were associated with an increased CVD risk. When the serum uric acid level was 6.9 mg/dL, the CVD risk was lowest. In women, a higher serum uric acid level was associated with an increased CVD risk.

As a pro-oxidant, uric acid can decrease the production and bioavailability of nitric oxide, activate NACHT, LRR and PYD domains-containing protein 3 inflammasome, and produce interleukin (IL)-1β, leading to activation of the renin−angiotensin system and endothelial dysfunction [2,9,19]. IL-1 production with IL-1 receptor activation is important in uric acid-related inflammation [2]. These actions of uric acid can increase CVD risk. The results of previous studies demonstrated a significant relationship between serum uric acid levels and CVD [8,9,10,20,21,22,23]. However, some studies showed that serum uric acid level was not an independent risk factor of CVD and not significantly associated with CVD mortality [11,12,24]. Different inclusion criteria such as age and comorbidities, the small number of study participants, confounding factors including the use of diuretics and urate-lowering agents, and the antioxidant effect of uric acid were suggested as reasons for refuting the independent relationship between serum uric acid levels and CVD [8,11,12].

In this study, we revealed that hyperuricemia was associated with an increased CVD risk. This result was obtained by analyzing the nationwide data representative of the general population of Korean adults. CVD risk was calculated on the basis of 10-year CVD risk scores using age, total cholesterol, HDL-C, systolic blood pressure, hypertension treatment status, smoking status, and diabetes, which were established risk factors of CVD. The association was adjusted for covariates in assessing CVD risk, including BMI, physical activity, serum creatinine, and alcohol consumption.

Recent studies have reported on the association between lower serum uric acid levels and CVD, showing an approximate U-shaped distribution [9,10,13,21]. These studies have demonstrated that lower serum uric acid levels were related to higher CVD mortality rates. One study on the risk of CVD by using urate-lowering agents suggested that CVD risk might be increased by an excessively low serum uric acid level [25]. Our findings also suggested an approximate U-shaped association in men and the lowest CVD risk at 6.9 mg/dL serum uric acid level. The mechanism of higher CVD risk related to lower serum uric acid levels has not been clearly explained in the previous studies. One of the explanations is the loss of antioxidant effect due to low serum uric acid. Uric acid has antioxidant and pro-oxidant effects. As a powerful antioxidant, uric acid scavenges oxygen radicals and protects LDL-C from oxidation, resulting in a protective effect against atherosclerosis [19]. Individuals with low serum uric acid levels are postulated to have an increased CVD risk because of the decreased antioxidant effect of serum uric acid [9,10,21,25]. A similar explanation was provided in a study of kidney function loss [26]. Renal hypouricemia decreases the ability to respond to oxidative stress, resulting in acute kidney injury. Another explanation is that a low serum uric acid level possibly reflects malnutrition [9,21]. A recent study showed that malnutrition was an important factor in higher CVD-related mortality in elderly individuals with hypouricemia [21]. Hypouricemia-related mortality got higher with more severe malnourishment in elderly patients.

Our study showed a sex−based difference in CVD risk in the presence of low serum uric acid levels. Men with lower serum uric acid levels had increased CVD risks, but women with lower serum uric acid levels had decreased CVD risks. Some studies showed a stronger association between hyperuricemia and CVD events but lower mortality rates from CVD at lower serum uric acid levels in women [8,13,27]. Most previous studies explained the sex-based difference related to the uric acid level in terms of the change in estrogen level in postmenopausal women [8,13,28,29]. Owing to the cardioprotective effect of estrogen, older women with decreased levels of estrogen tend to have higher risks of CVD. Another study showed that the mean age at higher serum uric acid levels was higher in women [13]. Our study also showed that women with high serum acid levels were older.

This study showed that aerobic physical activity was related to a decreased 10-year CVD risk, but BMI and serum creatinine were related to an increased risk. Alcohol consumption was related to a decreased 10-year CVD risk in men but increased risk in women. Among the lifestyle factors for the development of CVD, physical activity reportedly has an inverse relationship with CVD risk [30,31,32]. The WHO recommends that adults exercise for ≥75 min every week at a moderate or high intensity [33]. This study showed that participants who practiced aerobic physical activity at the recommended level had lower CVD risks than those who did not. Physical activity can control serum levels of HDL-C and LDL-C, and coronary artery calcium; improve insulin sensitivity, blood coagulability, and oxygenation to the myocardium; and decrease the activities of the renin−angiotensin system and sympathetic nervous system tone for lowering blood pressure [30,31,34]. BMI is also a lifestyle factor for the development of CVD [35,36]. More than 1 billion adults have been estimated by the WHO to be overweight, and >300 million adults are obese [35]. A high BMI is more likely to cause CVD at a younger age [37]. In this study, higher BMI was related to an increased CVD risk. Insulin resistance and low-grade inflammation are involved in the relationship between obesity and CVD risk [38]. Many studies have reported the association of chronic kidney disease (CKD) with CVD risk [39,40]. This study also showed that decreased kidney function was related to an increased 10-year CVD risk. Patients with CKD have some common risk factors of CVD, including hypertension, diabetes, and smoking. To prevent CVD among patients with CKD, the control of CVD risk factors, and the implementation of a strategy for inhibiting the progression of kidney dysfunction are recommended. The relationship between alcohol consumption and CVD is complex [41,42]. Although low or moderate alcohol consumption could slightly decrease the risk of CVD events, its protective effect varies with geographic regions or ethnicity, and higher alcohol consumption increases CVD risk. We think that in addition to the regulation of uric acid, management of modifiable factors such as aerobic physical activity, BMI, and alcohol consumption is also required to manage CVD risk.

This study has some limitations. It had a cross-sectional design, which limited our ability to assess causal relationships. We did not adjust for the education level and dietary factors that could change the serum uric acid level. Information about medications, including diuretics and urate-lowering agents, was not available. Nevertheless, our study has several strengths. This is the first population-based nationwide study to analyze the relationships of serum uric acid levels to CVD risk by using the Framingham risk score. We used big data collected using a standardized protocol from a nationally representative survey that included a large sample of the general population. We adjusted for various potential confounding factors. Our study could contribute to the development of a management strategy for hyperuricemia and determination of the target level of serum uric acid.

## 5. Conclusions

Hyperuricemia was associated with an increased CVD risk in both sexes and hypouricemia was associated with an increased CVD risk in men. We, therefore, recommend proper management of uric acid levels in the general population to reduce CVD risks.

## Figures and Tables

**Figure 1 ijerph-16-04632-f001:**
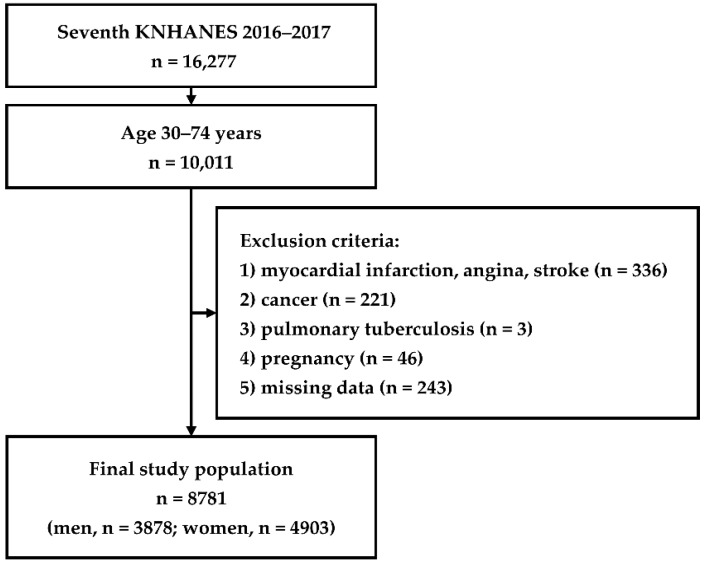
Flow chart showing the inclusion and exclusion criteria of the study

**Figure 2 ijerph-16-04632-f002:**
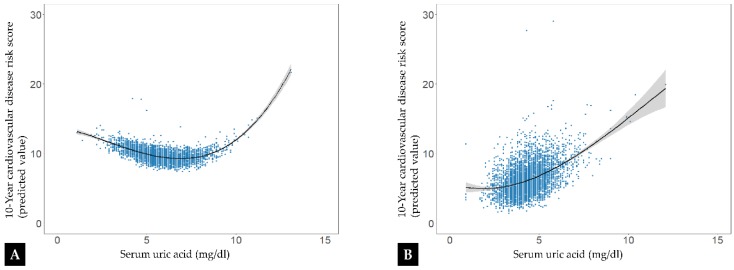
Associations of serum uric acid with cardiovascular disease risk (adjusted for aerobic physical activity, muscular strength exercise, body mass index, serum creatinine, and alcohol consumption) in men (**A**) and women (**B**).

**Table 1 ijerph-16-04632-t001:** General characteristics of the study participants.

Characteristics	Men (*n* = 3878)			Women (*n* = 4903)		
	Serum Uric Acid, mg/dL		*p* Value	Serum Uric Acid, mg/dL		*p* Value
	≥7 (*n* = 701)	<7 (*n* = 3177)		≥6 (*n* = 279)	<6 (*n* = 4624)	
Age, mean (S.D.), years	46.46 (11.08)	49.91 (11.66)	<0.001	54.15 (11.99)	49.80 (11.58)	<0.001
Education (%)			0.005			0.003
middle school	15.76	21.03		35.18	28.15	
high school	28.95	29.90		37.89	33.61	
college or higher	55.29	49.06		26.94	38.24	
Household income (%)			0.169			0.118
low	9.00	11.05		18.20	13.16	
middle	53.23	54.78		52.99	55.79	
high	37.77	34.17		28.80	31.06	
Smoking status (%)			0.186			0.057
non-current smoker	58.83	61.75		91.67	94.75	
current smoker	41.17	38.25		8.33	5.25	
Alcohol consumption (%)			0.001			0.045
<2 times per week	71.75	78.59		91.70	94.69	
≥2 times per week	28.25	21.41		8.30	5.31	
Hypertension status (%)			0.323			<0.001
no	78.38	80.08		67.38	84.92	
yes	21.62	19.92		32.62	15.08	
Diabetes status (%)			0.001			0.001
no	94.73	89.79		85.56	93.37	
yes	5.27	10.21		13.44	6.63	
Practice rate of aerobic physical activity (%)			0.077			0.472
yes	50.89	46.67		41.00	43.44	
no	49.11	53.33		59.00	56.56	
Practice rate of muscular strength exercise (%)			0.750			0.743
yes	25.53	26.19		15.64	14.73	
no	74.47	73.81		84.36	85.27	
Systolic blood pressure, mean (S.D.), mmHg	123.29 (14.15)	120.07 (14.28)	<0.001	121.38 (18.35)	115.22 (16.28)	<0.001
Body mass index, mean (S.D.), kg/m^2^	25.90 (3.47)	24.42 (3.07)	<0.001	25.96 (3.88)	23.41 (3.55)	<0.001
Total cholesterol, mg/dL	205.12 (39.04)	193.57 (37.03)	<0.001	206.41 (38.40)	196.99 (36.89)	<0.001
HDL-C, mg/dL	43.89 (9.69)	47.61 (11.48)	<0.001	50.80 (13.10)	55.13 (12.77)	<0.001
Serum creatinine, mean (S.D.), mg/dL	1.02 (0.25)	0.95 (0.33)	<0.001 *	0.85 (0.44)	0.70 (0.16)	<0.001 *

* Log transformed, HDL-C: high-density lipoprotein cholesterol.

**Table 2 ijerph-16-04632-t002:** Associations of serum uric acid and covariates with cardiovascular disease risk.

Parameters	Men			Women		
	*β*-Coefficient	SE	*p* Value	*β*-Coefficient	SE	*p* Value
Serum uric acid #1	0.023	0.003	<0.001	0.006	0.001	<0.001
Serum uric acid #2	−0.243	0.033	<0.001	0	0	0
Practice rate of aerobic physical activity	−1.067	0.213	<0.001	−0.827	0.168	<0.001
Practice rate of muscular strength exercise	0.452	0.234	0.055	−0.012	0.214	0.956
Body mass index	0.144	0.031	<0.001	0.517	0.031	<0.001
Serum creatinine	0.763	0.367	0.039	2.776	0.525	<0.001
Alcohol consumption	0.467	0.229	0.043	−1.620	0.390	<0.001

Serum uric acid #1 = serum uric acid^3^, Serum uric acid #2 = serum uric acid^2^, SE: standard error.

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
