# Peer review of "Association of Serum Uric Acid with Cardiovascular Disease Risk Scores in Koreans"

_ijerph, 2019, doi:10.3390/ijerph16234632_

Round 1

Reviewer 1 Report

This manuscript by Lee et al. focused on the association of serum uric acid, physical activity, and body mass index (BMI) with cardiovascular disease (CVD) risk. Authors demonstrated that hyperuricemia was associated with increased CVD risk, and hypouricemia was associated with increased CVD risk in men. In addition, aerobic physical activity and lower BMI were associated with reduced CVD risk. As authors mentioned, the relation between serum uric acid levels and CVD risk remains controversial. Therefore, the concept of this study is valuable and the results seem agreeable. Although overall manuscript seems written very well, authors may want to resolve several issues as below.

Major comments;

1) Although authors focused on the risk for CVD of not only serum uric acid levels but also aerobic physical activity and BMI, most of description was occupied with the results of serum uric acid levels in this manuscript. It seems better that authors focus on only serum uric acid levels in this manuscript. If authors want to describe multiple risk factors for CVD, other factors not only aerobic physical activity or BMI should be evaluated and discussed.

Minor comments;

1) If 10-year risk of CVD were evaluated, hazard ratio obtained from the Cox proportional hazard model should be used instead of odds ratio.

2) Authors may want to discuss more the reasons or mechanisms why hypouricemia was associated with higher risk of CVD in only men.

Author Response

Title “Associations of serum uric acid with cardiovascular disease risk scores in Koreans”

Although authors focused on the risk for CVD of not only serum uric acid levels but also aerobic physical activity and BMI, most of description was occupied with the results of serum uric acid levels in this manuscript. It seems better that authors focus on only serum uric acid levels in this manuscript. If authors want to describe multiple risk factors for CVD, other factors not only aerobic physical activity or BMI should be evaluated and discussed.

->We agree with your comment. Our primary aim was to identify the association between serum uric acid and CVD risk. Accordingly, we changed the title and clarified this purpose in the text (page 2, line 53−54).

: “Here we aimed to investigate the association between serum uric acid and CVD risk in a nationwide survey of general Korean adults using the 10-year CVD risk score.”

And we made an additional adjustment of other covariates including serum creatinine and alcohol consumption. We added this result in the text including Table 2 (page 4, line 152−162 and 165−168).

: “The relationships of serum uric acid, aerobic physical activity, BMI, serum creatinine, and alcohol consumption with CVD risk were statistically significant. In both men and women, the 10-year CVD risk score, aerobic physical activity, and alcohol consumption in women were negatively correlated but the 10-year CVD risk score and BMI, serum creatinine, and alcohol consumption in men were positively correlated. The 10-year CVD risk score and muscular strength exercise had no significant relationship. Figure 2 presents the association between serum uric acid and the 10-year CVD risk analyzed by fractional polynomial model. In the case of men, the third-degree equation cubic fraction was fitted, as follows: 10-year CVD risk score = 0.023 × serum uric acid3 – 0.243 × serum uric acid2 – 1.067 × practice rate of aerobic physical activity + 0.452 × practice rate of muscular strength exercise + 0.144 × BMI + 0.763 × serum creatinine + 0.467 × alcohol consumption (root mean square error [RMSE] = 5.3714).”

: “Also, in the case of women, the third-degree equation cubic fraction was fitted: 10-year CVD risk score = 0.006 × serum uric acid3 – 0.827 × practice rate of aerobic physical activity – 0.012 × practice rate of muscular strength exercise + 0.517 × BMI + 2.776 × serum creatinine – 1.620 × alcohol consumption (RMSE = 5.4784).”

If 10-year risk of CVD were evaluated, hazard ratio obtained from the Cox proportional hazard model should be used instead of odds ratio.

->We assessed CVD risk using the 10-year CVD risk score, which is not a follow-up result over 10 years but a value predicted by a certain formula (page 3, line 105−111).

: “The formula was developed using a sex-specific multivariate risk factor algorithm based on the Framingham Heart Study. This scoring model is intended for participants free of CVD in a baseline examination and aged 30–74 years. This formula has been validated in large-population studies and allows the screening of high-risk candidates among the general population for initial CVD events. The 10-year CVD risk score is the calculated likelihood of CVD after 10 years based on age, total cholesterol, HDL-C, systolic blood pressure, hypertension treatment status, smoking status, and diabetes.”

The data examined in this study were collected through the cross-sectional survey in 2016 and 2017 for the same items and variables. Due to the nature of the data, we were able to calculate odds ratios, not hazard ratios.

Authors may want to discuss more the reasons or mechanisms why hypouricemia was associated with higher risk of CVD in only men.

->As you pointed out, when serum uric acid level decreased below a certain level, CVD risk in men increased while that in women gradually decreased and remained constant. Some studies showed that women have a stronger association with CVD at high serum uric acid level but a lower mortality rate from CVD at low serum uric acid level. A thorough search of previous studies suggested that estrogen and age appeared to be the cause. For this part we have added some explanation in the text (page 7, line 222−224 and 225−229).

: “Some studies showed a stronger association between hyperuricemia and CVD events while a lower mortality rate from CVD at low serum uric acid level in women.”

: “Because of cardioprotective effect of estrogen, older women with decreased level of estrogen tend to have a higher risk of CVD. Another possible explanation by one study was that the mean age at higher serum uric acid level was higher in women. Our study also showed that women were older in high serum acid group.”

Reviewer 2 Report

1) The analyses regarding physical activity and BMI are poorly developed and distracting. Rather, they should be considered as covariates for the primary analysis of interest (uric acid and CVD score). Other covariates that are displayed in Table 1 should also be considered in adjusted models for this primary analaysis. 2) The subanalysis of men with low uric acid in Table 3 is not justified. Why was this group studied and not others? Why were simple unadjusted odds ratios calculated (and what are the units/interpretation of this odds ratio). This section of the manuscript is also poorly developed and should be removed or greatly expanded (justification for this analysis, statistical methods improved and better explained).

Author Response

Title “Associations of serum uric acid with cardiovascular disease risk scores in Koreans”

The analyses regarding physical activity and BMI are poorly developed and distracting. Rather, they should be considered as covariates for the primary analysis of interest (uric acid and CVD score).

->Thank you for pointing out this important issue. Our primary aim was to identify the association between serum uric acid and CVD risk. In response to your comment, we regarded physical activity and BMI as covariates for our primary analysis and made an additional adjustment of other covariates including serum creatinine and alcohol consumption. This modification was added in the text including Table 2 (page 4, line 136−138, page 4, line 152−162 and 165−168).

: “After the adjustment for serum uric acid, aerobic physical activity, muscular strength exercise, body mass index (BMI), serum creatinine, and alcohol consumption, the β-coefficients between these variables and 10-year CVD risk score were calculated.”

: “The relationships of serum uric acid, aerobic physical activity, BMI, serum creatinine, and alcohol consumption with CVD risk were statistically significant. In both men and women, the 10-year CVD risk score, aerobic physical activity, and alcohol consumption in women were negatively correlated but the 10-year CVD risk score and BMI, serum creatinine, and alcohol consumption in men were positively correlated. The 10-year CVD risk score and muscular strength exercise had no significant relationship. Figure 2 presents the association between serum uric acid and the 10-year CVD risk analyzed by fractional polynomial model. In the case of men, the third-degree equation cubic fraction was fitted, as follows: 10-year CVD risk score = 0.023 × serum uric acid3 – 0.243 × serum uric acid2 – 1.067 × practice rate of aerobic physical activity + 0.452 × practice rate of muscular strength exercise + 0.144 × BMI + 0.763 × serum creatinine + 0.467 × alcohol consumption (root mean square error [RMSE] = 5.3714).”

: “Also, in the case of women, the third-degree equation cubic fraction was fitted: 10-year CVD risk score = 0.006 × serum uric acid3 – 0.827 × practice rate of aerobic physical activity – 0.012 × practice rate of muscular strength exercise + 0.517 × BMI + 2.776 × serum creatinine – 1.620 × alcohol consumption (RMSE = 5.4784).”

Other covariates that are displayed in Table 1 should also be considered in adjusted models for this primary analysis.

->The main purpose of our study was to analyze the relationship between serum uric acid and 10-year CVD risk. Among variables with significant P value in Table 1, sex, age, status of hypertension and diabetes, systolic blood pressure, total cholesterol, and HDL-cholesterol were not adjusted because these had been already reflected in 10-year CVD risk score calculation. The remaining variables with significant P value including alcohol consumption, BMI, and serum creatinine were adjusted to be analyzed in Table 2. Although not significant, aerobic physical activity and muscular strength exercise were adjusted at the discretion of researchers. For distraction due to unadjusted education level, the description of the text was clarified and added at the limitations.

The subanalysis of men with low uric acid in Table 3 is not justified. Why was this group studied and not others? Why were simple unadjusted odds ratios calculated (and what are the units/interpretation of this odds ratio). This section of the manuscript is also poorly developed and should be removed or greatly expanded (justification for this analysis, statistical methods improved and better explained).

->Our original intention was to identify the difference between high and low groups of 10-year CVD risk in male with a serum uric acid level < 6.9 mg/dl through the analysis of Table 3. As shown in Figure 2A, 10-year CVD risk increased as the serum uric acid decreased below a certain level. We then wanted to investigate how muscular strength exercise, aerobic physical activity, and BMI affect CVD risk in the group with low serum uric acid in addition to established risk factors for CVD including age, smoking, hypertension, diabetes, and blood pressure. On the other hand, since risk factors such as age and smoking were already included in the 10-year CVD risk prediction formula based on Framingham Heart Study findings, they could not be adjusted again, so we were forced to calculate the unadjusted odds ratio.

However, as you pointed out, there are insufficient statistical methods and inadequate model settings, we deleted “Table 3” and subsection 3.3 and the text related to them.
